# Volcanic Eruption of Cumbre Vieja, La Palma, Spain: A First Insight to the Particulate Matter Injected in the Troposphere

Michaël Sicard [1,2,*], Carmen Córdoba-Jabonero [3], Africa Barreto [4], Ellsworth J. Welton [5], Cristina Gil-Díaz [1], Clara V. Carvajal-Pérez [3], Adolfo Comerón [1], Omaira García [4], Rosa García [6], María-Ángeles López-Cayuela [3], Constantino Muñoz-Porcar [1], Natalia Prats [4], Ramón Ramos [4], Alejandro Rodríguez-Gómez [1], Carlos Toledano [7] and Carlos Torres [4]

1   CommSensLab, Department of Signal Theory and Communications, Universitat Politècnica de Catalunya, 08034 Barcelona, Spain; cristina.gil.diaz@upc.edu (C.G.-D.); adolfo.comeron@upc.edu (A.C.); constantino.munoz@upc.edu (C.M.-P.); alejandro.rodriguez.gomez@upc.edu (A.R.-G.)
2   Ciències i Tecnologies de l'Espai-Centre de Recerca de l'Aeronàutica i de l'Espai/Institut d'Estudis Espacials de Catalunya (CTE-CRAE/IEEC), Universitat Politècnica de Catalunya, 08034 Barcelona, Spain
3   Atmospheric Research and Instrumentation Branch, Instituto Nacional de Técnica Aeroespacial (INTA), 28850 Torrejon de Ardoz, Spain; cordobajc@inta.es (C.C.-J.); ccarper@inta.es (C.V.C.-P.); lopezcma@inta.es (M.-Á.L.-C.)
4   Izaña Atmospheric Research Center, State Meteorological Agency of Spain (AEMET), 38001 Santa Cruz de Tenerife, Spain; abarretov@aemet.es (A.B.); ogarciar@aemet.es (O.G.); npratsp@aemet.es (N.P.); rramosl@aemet.es (R.R.); ctorresg@aemet.es (C.T.)
5   Code 612, Goddard Space Flight Center, National Aeronautics and Space Administration, Greenbelt, MD 20771, USA; ellsworth.j.welton@nasa.gov
6   TRAGSATEC, 28006 Madrid, Spain; rgarci47@tragsa.es
7   Group of Atmospheric Optics, Universidad de Valladolid, 47011 Valladolid, Spain; toledano@goa.uva.es
*   Correspondence: michael.sicard@upc.edu; Tel.: +34-934-011-065

**Abstract:** The volcanic eruption of Cumbre Vieja (La Palma Island, Spain), started on 19 September 2021 and was declared terminated on 25 December 2021. A complete set of aerosol measurements were deployed around the volcano within the first month of the eruptive activity. This paper describes the results of the observations made at Tazacorte on the west bank of the island where a polarized micro-pulse lidar was deployed. The analyzed two-and-a-half months (16 October–31 December) reveal that the peak height of the lowermost and strongest volcanic plume did not exceed 3 km (the mean of the hourly values is $1.43 \pm 0.45$ km over the whole period) and was highly variable. The peak height of the lowermost volcanic plume steadily increased until week 11 after the eruption started (and 3 weeks before its end) and started decreasing afterward. The ash mass concentration was assessed with a method based on the polarization capability of the instrument. Two days with a high ash load were selected: The ash backscatter coefficient, aerosol optical depth, and the volume and particle depolarization ratios were, respectively, 3.6 (2.4) $Mm^{-1}sr^{-1}$, 0.52 (0.19), 0.13 (0.07) and 0.23 (0.13) on 18 October (15 November). Considering the limitation of current remote sensing techniques to detect large-to-giant particles, the ash mass concentration on the day with the highest ash load (18 October) was estimated to have peaked in the range of 800–3200 $\mu g\, m^{-3}$ in the lowermost layer below 2.5 km.

**Keywords:** Cumbre Vieja; volcano; eruption; fresh ash particles; remote sensing

## 1. Introduction

Between 11 and 19 September 2021, the island of La Palma, in the Spanish archipelago of the Canary Islands, experienced an intense seismic swarm, with more than 6000 recorded earthquakes [1], which culminated on 19 September with the start of the eruption of the Cumbre Vieja volcano [2]. The volcano eruption cost the life of one victim, the injection of ash in the atmosphere forced the evacuation of residents and the cancelation of hundreds

of flights, and pulsating fountains of lava destroyed infrastructures and partially damaged La Palma's banana crop [3]. From the very first day, the volcano activity was monitored by the Plan de Emergencias Volcánicas de Canarias (PEVOLCA), which emitted daily reports available in [4]. These reports were performed mostly on the basis of ground-based seismic data, sulfur dioxide ($SO_2$), carbon dioxide ($CO_2$), and $PM_{10}$ concentrations, as well as satellite images. The eruptive process of solid material finished on 13 December, and the eruption of Cumbre Vieja was declared terminated on 25 December [4].

As an unprecedented collaborative scientific effort in the framework of Aerosol, Clouds, and Trace Gases Research Infrastructure (ACTRIS) and ACTRIS-Spain [5], coordinated by the State Meteorological Agency of Spain (AEMET), several research groups and private organizations across Europe sent scientific instrumentation to the area of the eruption to study the characteristics of the volcanic aerosols freshly emitted in the atmosphere. This paper presents the first results obtained from the micro-pulse lidar system observations performed starting on 16 October 2021 at Tazacorte (28.641°N, 17.933°W, 140 m asl) on the west bank of the island. The observation site location, together with the region of the volcano's most active vents, is depicted in Figure 1. The temporal evolution of the lowermost volcanic layer height is first analyzed, and the retrieval of the ash and non-ash mass concentration vertical profiles in the troposphere is discussed in two selected case studies.

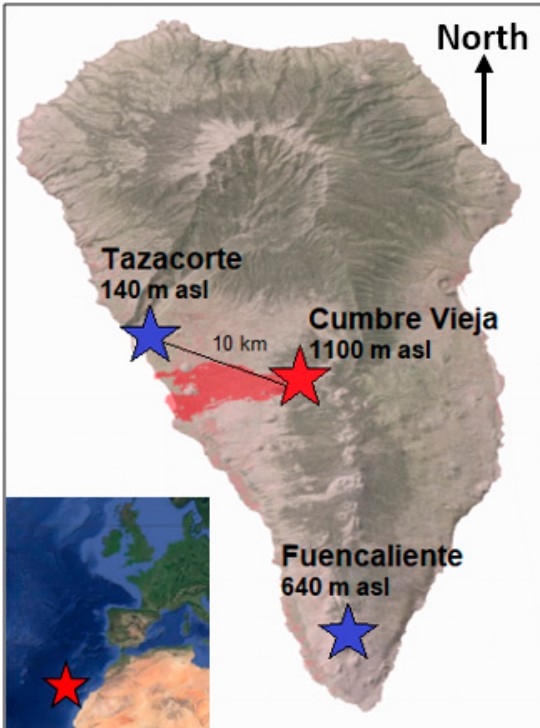

**Figure 1.** Geographical position of Cumbre Vieja volcano and Tazacorte site in La Palma Island. The red-shaded area represents the extension of the area affected by the lava flow. Credits: https://ign-esp.maps.arcgis.com/ (last accessed on 19 January 2022). Insert from Google Maps.

## 2. Materials and Methods

### 2.1. Instrumentation

The two instruments used intensively in this study were deployed specifically to monitor the volcano eruption:

1.  A micro-pulse lidar (MPL, model MPL-4B) was operated in Tazacorte from 15 October 2021 to 25 January 2022. This instrument is an eye-safe elastic lidar operating at 532 nm, with depolarization capability [6]. It operates continuously (24/7) with a low-pulse energy (5–6 µJ) laser and a repetition rate of 2.5 kHz. A dead-time correction was

applied, following the manufacturer's instructions and laboratory calibrations of the detector [7]. Dark-count and after-pulse measurements were performed monthly [7,8]. The overlap correction was performed, with an overlap estimation performed prior to shipping of the instrument to La Palma Island and according to [9,10]. Two fans were located next to the output window to continuously blow away the ash that was constantly deposited on it. The inversion of the MPL signals provides the vertical profile of the aerosol backscatter coefficient, the volume depolarization ratio, and the particle depolarization ratio.

2. A sun–sky photometer was operated in Fuencaliente (28.487°N, 17.849°W, 630 m asl, 21 km southeast of Tazacorte) from 21 September 2021 to 23 January 2022. Data from this instrument are available at the "La Palma" site of the Aerosol Robotic Network (AERONET, https://aeronet.gsfc.nasa.gov/; last accessed 17 January 2022). This instrument was used to estimate the mass conversion factors used in Section 2.2. Inversion products Level 1.5 version 3 [11] were used. Preliminary aerosol vertical profiles using a synergy between this instrument and a ceilometer in Fuencaliente are presented in [12].

Additional devices deployed in Tazacorte by the AEMET were used occasionally to deepen our analysis—namely, a pulsed fluorescence Thermo Teco analyzer ($SO_2$ concentration) [13] and an optical particle counter GRIMM 1.108 ($PM_{2.5}$ and $PM_{10}$) [14].

### 2.2. Methodology

The methodology concerns essentially the processing applied to the MPL data, which was performed exclusively with in-house algorithms. The considered period lasted from 16 October to 31 December 2021. No volcanic plume as such was detected in January 2022.

### 2.2.1. Profile Screening

Profiles integrated over one hour were used to retrieve the height of the peak of the lowermost volcanic plume. This plume usually originates directly from the volcano, while upper plumes (when present) are more likely due to sporadic, short boost of the explosive activity or recirculation. Here, recirculation refers to the atmospheric transport of upper volcanic plumes by prevailing winds out of the Island and the subsequent transport onshore of buoyant air due to local breeze patterns. In general, the lowermost volcanic plume showed a peak (a local maximum, $h_P$) immediately at the top of the marine boundary layer (MBL). Due to the proximity of the site to the volcano (10 km), the volcanic plume was often so dense that it blocked the lidar signal. In order to avoid the cases for which the laser light could not penetrate the lowermost volcanic plume, which would bias negatively the retrieval, a simple threshold method [15] was applied to the range-square-corrected lidar signal (RSCS) calculated, following [16]. The RSCS at 8 km was taken as the reference value. If it was lower than 0.02 (arbitrary units), the profile was classified as blocked or strongly attenuated, otherwise, it was classified as not blocked. The value of 0.02 was empirically determined. We have checked that a RSCS value of 0.02 corresponds approximately to a signal-to-noise ratio of 10 during daytime (>10 during nighttime).

### 2.2.2. Separation of Ash and Non-Ash Particles

As far as volcanic matter concentration is concerned, we applied the Polarization Lidar Photometer Networking (POLIPHON) algorithm in order to separate the fine- and coarse-mode components and estimate their mass concentration [17]. POLIPHON was applied successfully to MPL measurements for different aerosol mixtures [18]. The method is based on the attribution of specific particle linear depolarization ratios ($\delta$), lidar ratio ($S$), and mass conversion factors ($\overline{\nu/\tau} \cdot \rho$, i.e., volume-to-AOD ratio times particle density), for both mode components. In the case of volcanic aerosols, the fine mode represents mainly non-ash particles including sulfur dioxide, while the coarse mode represents ash particles. The steps are as follows:

1. Separation of the MPL total backscatter coefficient ($\beta$) into non-ash and ash modes;

2.  Calculation of the mass concentration ($m$) for both modes as $m = \beta \cdot S \cdot \overline{v/\tau} \cdot \rho$.

The specific values of $\delta$, $S$, and $\rho$ for non-ash and ash ($n$-$a$ and $a$ suffixes, respectively) were taken from [17]: $\delta_{n-a} = 0.01$ and $\delta_a = 0.36$; $S_{n-a} = S_a = 50$ sr; $\rho_{n-a} = 1.5$ and $\rho_a = 2.6$ g cm$^{-3}$. $\overline{v/\tau}$ for non-ash and ash concentrations were computed from, respectively, fine- and coarse-mode AERONET products of column volume concentration ($v$) and AOD ($\tau$) following [17]. The overline of $\overline{v/\tau}$ indicates that it is a column-equivalent ratio, independent of the height, contrarily to $\beta$.

## 3. Results, Discussion, and Conclusions

### 3.1. Temporal Evolution 16 October–31 December 2021

Figure 2 shows the time–height plot of the volume depolarization ratio, $\delta^V$, and the temporal evolution of the peak height, $h_P$, of the lowermost volcanic plume. Weekly mean values of $h_P$ are given in Table 1. The color code in Figure 2a was adjusted so that black is representative of the molecular level. For the time–height plot, $\delta^V$ was preferred against $\delta$ because of its physical meaning even in a molecular atmosphere and a higher signal-to-noise ratio. Luckily, very few mineral dust intrusions occurred during the period under study. Two were clearly identified as such: one between 19 October 00 UTC and 22 October 11 UTC (first marked dust event in the figure) and another one after the end of the eruption (second marked dust event in the figure). The 1-day average of the peak height of the lowermost volcanic plume did not exceed 3 km, and lofted layers (>3 km) were relatively seldom.

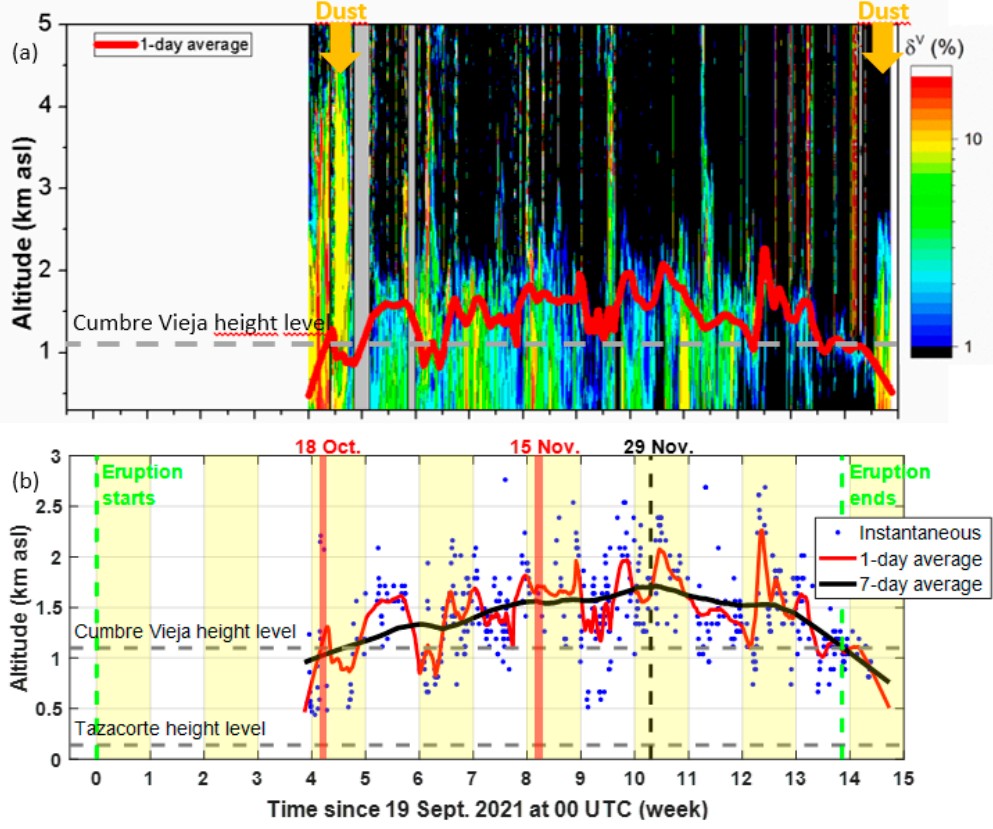

**Figure 2.** (a) Time–height plot of the volume depolarization ratio, $\delta^V$, and (b) temporal evolution of the peak height of the lowermost volcanic plume, $h_P$. The green dashed vertical lines indicate the start and end of the eruption. The black dashed vertical line indicates the gradient switch of the 7-day average from positive to negative. The two red-shaded areas indicate the days discussed in Section 3.2.

**Table 1.** Weekly means ($\mu$) and standard deviations ($\sigma$) of $h_P$.

| Week | | 4 | 5 | 6 | 7 | 8 | 9 | 10 | 11 | 12 | 13 | 14 | 15 | 4–15 |
|---|---|---|---|---|---|---|---|---|---|---|---|---|---|---|
| $h_p$ (km) | $\mu$ | 0.71 | 1.04 | 1.59 | 1.27 | 1.46 | 1.72 | 1.55 | 1.76 | 1.41 | 1.60 | 1.22 | 0.98 | 1.43 |
| | $\sigma$ | 0.26 | 0.53 | 0.18 | 0.38 | 0.32 | 0.41 | 0.51 | 0.38 | 0.36 | 0.50 | 0.32 | 0.13 | 0.45 |

It is worth noting that the gaps in the instantaneous (hourly) values (Figure 2b) were produced by the profile screening applied (Section 2.2.1): Among all available 1 h time-resolution profiles, 62% were classified as not blocked, meaning that 38% of the time, the lowermost volcanic plume was optically so thick that the laser light could not penetrate it. In most cases, $h_P$ varied between 0.5 and 2.5 km, i.e., in an interval including the MBL height estimated in Tenerife (where the MBL behavior is comparable to that at Tazacorte in the absence of mineral dust), varying between 1.3 and 1.6 km [19]. Indeed, the average of $h_P$ over the whole period was $1.43 \pm 0.45$ km. This result suggests that the peak of the lowermost volcanic plume was, in general, slightly above the volcano altitude (1.1 km); therefore, the largest concentrations of volcanic matter at Tazacorte should have been expected not at ground level but in the bottom part of the troposphere. However, some periods with fast sinking ($h_P \simeq 0.5$ km, i.e., well below the volcano altitude) were observed on weeks 5 and 10. On week 5, this sinking of the lowermost volcanic plume was associated with a weekly peak of $SO_2$ and $PM_{2.5-10}$ at the surface [20]. For comparison, the authors of [21], who estimated the volcanic plume height of the eruption of Eyjafjallajökull volcano using a C-band weather radar, found a volcanic plume height at an altitude ranging from 5 to 10 km during explosive phases and generally in the first kilometer above the volcano altitude the rest of the time. In La Palma, the lowermost (usually coinciding with the strongest) volcanic plume was always observed at an altitude lower than 3 km. In such conditions, and leaving apart the periods when the lidar signal was blocked, our study revealed that the explosive phases of Cumbre Vieja were less intense than those of other recent volcanic eruptions. This was somehow confirmed by the rating on the volcanic explosivity index assigned by PEVOLCA, which was not higher than three (on a scale of eight) [4].

Considerable variability was observed in the instantaneous and daily means (Figure 2b), which is also reflected by the high standard deviations in Table 1. This result, together with the fact that Tazacorte was most of the time near to downwind of the volcanic plume dispersion with dominating northeast trade winds, suggested that the height of the lowermost volcanic plume was driven jointly by the dynamic of the volcano eruptive activity and the mesoscale meteorology.

*3.2. Ash Concentration Retrieval in the Troposphere*

The method described in Section 2.2.2 was applied to two case studies selected for potentially high ash concentrations in the atmospheric column. Selection criteria were high $PM_{2.5-10}$ concentrations, high AOD, and a non-attenuated lidar signal for the inversion to be reliable. $PM_{2.5-10}$ was used as an indicator of the presence of coarser particles—namely, ash in our study. Such conditions were found on 18 October, when $PM_{2.5-10}$ (AOD at 500 nm) daily mean was 37 µg m$^{-3}$ (0.31), and on 15 November, when $PM_{2.5-10}$ (AOD at 500 nm) daily mean was 38 µg m$^{-3}$ (0.36). Both days are identified in Figure 2b, and the TROPOMI $SO_2$ column concentration is shown in Figure 3. Although both daily means of $PM_{2.5-10}$ and AOD at 500 nm were similar on both days, a larger amount of $SO_2$ seemes to have been injected on 18 October based on the column data. Very scarce data on the particulate matter in La Palma Island are available in the literature for comparison. In one study [22], the authors reported long-term $PM_{10}$ measurements made in a regional background site in the Tenerife Island (comparable to Tazacorte) and found annual means of $PM_{10}$ of 19 µg m$^{-3}$. Long-term AOD measurements in Santa Cruz de Tenerife (52 m asl, Tenerife Island) were performed by [23] and, excluding mineral dust intrusions, revealed an annual mean AOD lesser than 0.15. Both $PM_{2.5-10}$ and AOD at 500 nm measured on 18

October and 15 November were approximately double these regional background values. Figure 4 shows hourly PM and $SO_2$ measurements for both days, as well as the profiles of the total, ash, and non-ash values of $\beta$ and $m$ for a selected time (03 UTC on 18 October and 15 UTC on 15 November). The profile of $\delta$ is also reported.

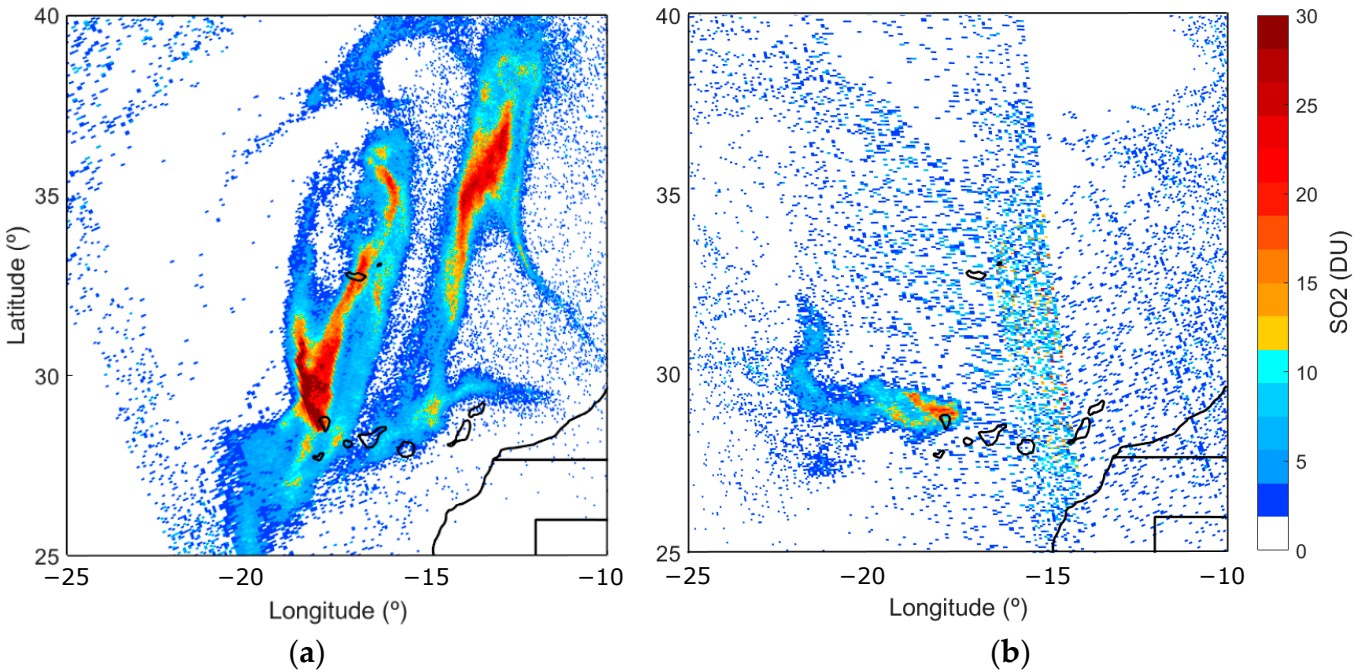

**Figure 3.** TROPOMI sulfur dioxide total vertical column (DU) over the Canary Islands on (**a**) 18 October and (**b**) 15 November 2021. Overpass time is about 13:30 local time.

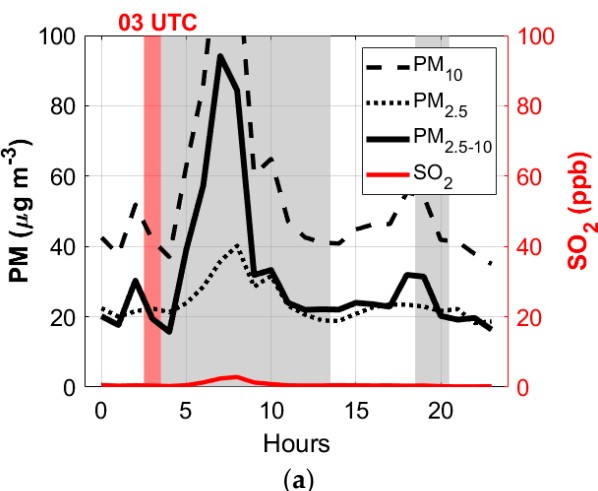
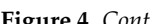
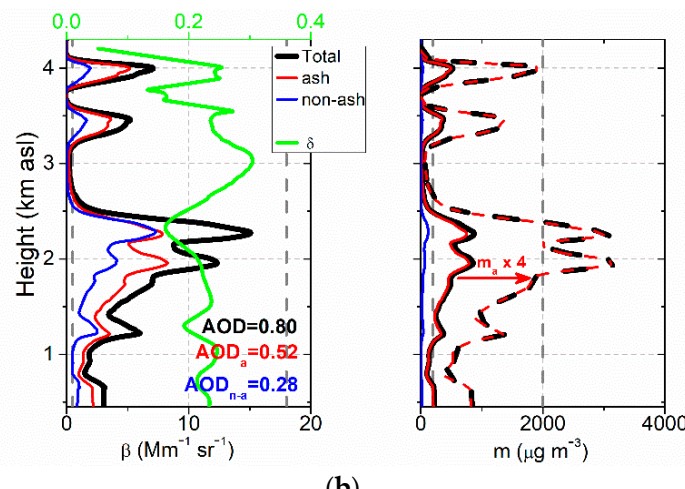

**Figure 4.** *Cont.*

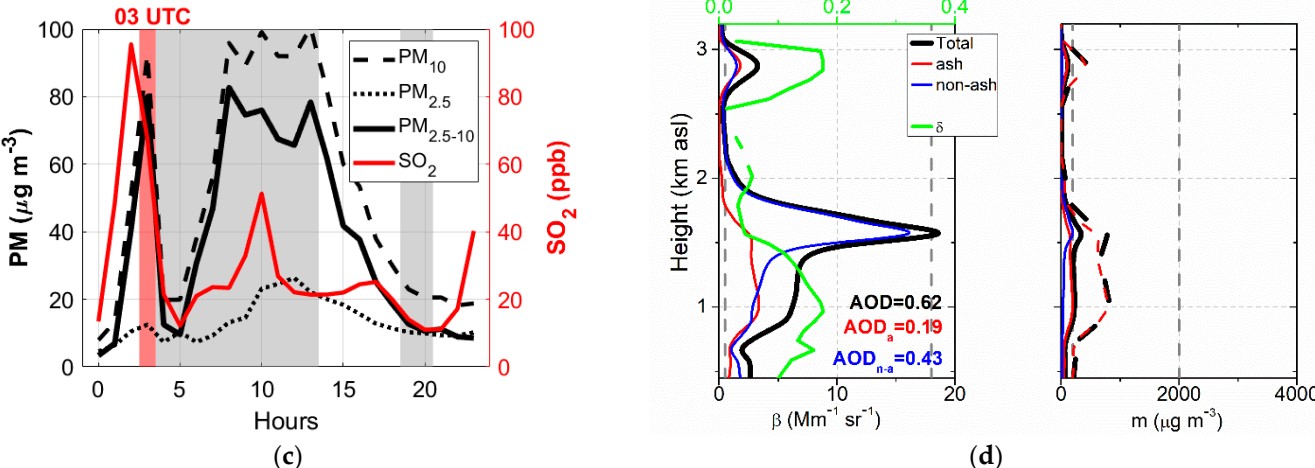

**Figure 4.** (**a**) Hourly PM and SO$_2$ concentrations on (**a**) 18 October and (**c**) 15 November. Grey areas indicate periods when the lidar signal was blocked. Profiles of total, ash, and non-ash particle backscatter coefficient and mass concentration on (**b**) 18 October at 03 UTC and (**d**) 15 November at 15 UTC. In the left plots, the particle depolarization ratio is also presented (upper axis), as well as the $\delta_{n-a}$ and $\delta_a$ values (grey, dash lines). In the right plot, the worst-case-scenario concentrations for total and ash mass concentration (Section 3.2) are represented by the dashed red and black curves.

### 3.2.1. 18 October at 03 UTC

Several volcanic layers were observed on the profiles of $\beta$ (Figure 4b): the lowermost layer up to 2.5 km and two thinner layers at 3.4 and 4.0 km. The total backscatter coefficient reached extremely large values, as high as 15 Mm$^{-1}$sr$^{-1}$. In most of the profiles, $\delta$ was high and varied between 0.2 and 0.3, indicating a strong contribution of the ash vs. non-ash particles in the whole profile. The total, ash and non-ash AOD were 0.80, 0.52, and 0.28, respectively. The ash AOD represented 65% of the total AOD. The non-ash AOD, 0.28, was far from being negligible. This result contrasted strongly with that of the surface SO$_2$, which was close to zero on that day (Figure 4a).

In order to calculate the mass concentration from the backscatter coefficient, the daily means of the instantaneous ratios at the Fuencaliente AERONET site were used: $\left(\overline{\nu/\tau}\right)_a = 0.736 \times 10^{-6}$ and $\left(\overline{\nu/\tau}\right)_{n-a} = 0.228 \times 10^{-6}$ m. These values were 22% and 29% larger than the values Ansmann et al. (2011) observed for a volcanic plume transported for 2–3 days. This is an expected result, as large-to-giant particles, expected to be present in fresh volcanic plumes, produce an almost linear increase in the coarse-mode $\overline{\nu/\tau}$ [24,25]. However, particles with radii larger than 15 μm are neglected in the AERONET inversion algorithm, which produces an underestimation of the coarse-mode volume concentration $\nu$ and, thus, of $\overline{\nu/\tau}$. In order to consider this potential limitation, we also computed the coarse-mode $\overline{\nu/\tau}$ for an assumed maximum effective coarse-mode radius ($r_{eff}$) of 4 μm [24], following the linear relationship $\overline{\nu/\tau} = 2r_{eff}/3$. proposed by [17] for size distributions dominated by large-to-giant particles. Then, the maximum $\overline{\nu/\tau}$ was estimated to be $2.7 \times 10^{-6}$ m, i.e., 3.7 times larger than the AERONET coarse-mode $\overline{\nu/\tau}$. To be on the safe side, the worst-case-scenario coarse-mode concentration was obtained by multiplying the AERONET coarse-mode $\overline{\nu/\tau}$ by 4. Two new profiles of the worst-case-scenario total and ash mass concentrations are reported in Figure 4b with dashed lines; they are indistinguishable because $m_a \gg m_{n-a}$. For reference, the limits of 200 and 2000 μg m$^{-3}$ are also reported in the figure. The first limit corresponds to the first contamination level defined by the UK Meteorological Office after the eruption of Eyjafjallajökull in 2010, and the second one is considered the maximum tolerable concentration for continuous flight operation [24]. $m_{n-a}$ was less than 200 μg m$^{-3}$ at all heights and almost a factor 10 smaller than $m_a$. In terms of mass integrated in the column, the ash particles contributed to more than 90% of the total mass. $m_a$ exceeded 200 μg m$^{-3}$ in the range 1.0–2.5 km and in the volcanic plumes at 3.4 and

4.0 km (peak of 795 $\mu$g m$^{-3}$ at 1.95 km). In the worst-case scenario, $m_a$ exceeded 200 $\mu$g m$^{-3}$ at all heights where the ash was present. The 2000-$\mu$g m$^{-3}$ limit was only exceeded in the worst-case scenario between 1.8 and 2.4 km (peak at 3180 $\mu$g m$^{-3}$). Our first estimation of $m_a$ was on the same order of magnitude as that of the Eyjafjallajökull plume observed over northern [17] (<1750 $\mu$g m$^{-3}$) and southern [25] (<1800 $\mu$g m$^{-3}$) Germany after 2–3 days of transport, and it was greater than that observed by [26] (80 $\mu$g m$^{-3}$) over Madrid after 4 days of transport or by [27] (180 $\mu$g m$^{-3}$) over Granada after 4–5 days of transport. Our results were also on the same order of magnitude as the mass concentration estimated by [28] (2000 $\mu$g m$^{-3}$) in a fresh volcanic plume of Etna volcano on 3 December 2015. However, the $m_a$ value in our worst-case scenario was smaller than the estimations made by [29] (>4000 $\mu$g m$^{-3}$) and [30] (24,000 $\mu$g m$^{-3}$) at the source in a fresh volcanic plume of Etna volcano on 15 November 2010. In that sense, and regarding the proximity of the site to the volcano (10 km), the amount of ash matter injected into the troposphere by Cumbre Vieja was estimated to be moderate. However, the 15 $\mu$m cutoff effect in the AERONET retrieval introduced uncertainty for the study of freshly emitted volcanic plumes dominated by large-to-giant particles, which studies of long-range transport plumes do not have to address [17,31].

### 3.2.2. 15 November at 15 UTC

On 15 November, the situation in the column is quite different, although the initial indicators of surface PM$_{2.5-10}$ and AOD were similar to those of 18 October. On 15 November at 15 UTC, two main volcanic layers are visible on the profiles of $\beta$ (Figure 4d): the lowermost layer up to 2.0 km and a thinner layer at 2.8 km. The total backscatter coefficient reached again extremely large values, as high as ~20 Mm$^{-1}$sr$^{-1}$ in the lowermost plume. $\delta$ was lower than on 18 October: It varied between 0.1 and 0.2, indicating a more equilibrated contribution of both ash and non-ash particles. At 15 UTC, the surface PM$_{2.5-10}$ concentration was 42 $\mu$g m$^{-3}$ (more than double the amount observed in the first case study) and the SO$_2$ concentration is 22 ppb (Figure 4c). While the SO$_2$ level was close to the average observed during the eruption (21 $\pm$ 39 ppb), the PM$_{2.5-10}$ level was almost three times larger than its average (15 $\pm$ 16 $\mu$g m$^{-3}$). The total, ash and non-ash AOD were 0.62, 0.19, and 0.43, respectively. $AOD_{n-a}$ represented 69 % of the total AOD. In the column, non-ash particles dominated, while at the surface, an exceptionally high amount of PM$_{2.5-10}$ (an indicator of coarser ash particles) was observed. The predominance of non-ash particles in the column was produced by the layer at 1.6 km. While $\beta_{n-a} \simeq \beta_a$ below 1.2 km and in the layer at 2.8 km, and $\beta_{n-a} \gg \beta_a$ around the peak at 1.6 km ($\delta \ll \delta_a$). We concluded that this strong layer was mostly made of non-ash particles and that this case was more likely representative of a more gaseous volcanic plume.

As far as mass concentration is concerned, the lower values of $\delta$ observed on 15 November at 15 UTC had a direct incidence on the mass retrieval of both components. On 15 November, $\left(\overline{\nu/\tau}\right)_a = 0.441 \times 10^{-6}$ and $\left(\overline{\nu/\tau}\right)_{n-a} = 0.163 \times 10^{-6}$ m. The fact that these volume-to-AOD ratios were smaller than those reported on 18 October might suggest that the particles of both components were smaller. This hypothesis deserves further investigation. $m_a$ still dominated at all altitudes, except in the layer at 1.6 km, where $m_a \simeq m_{n-a}$. Our first estimation of $m_a$ did not exceed 200 $\mu$g m$^{-3}$ (peak of 196 $\mu$g m$^{-3}$ at 1.04 km), so in the worst-case scenario ($m_a \times 4$), it did not exceed 800 $\mu$g m$^{-3}$ and stayed well below the 2000 $\mu$g m$^{-3}$ limit for continuous flight operation.

**Author Contributions:** Conceptualization, M.S.; methodology, M.S. and C.C.-J.; software, C.C.-J. and E.J.W.; formal analysis, M.S., C.C.-J. and C.G.-D.; investigation, M.S., A.B., E.J.W., C.V.C.-P., M.-Á.L.-C., R.G., C.M.-P. and A.R.-G.; resources, M.S., A.B., O.G., N.P., R.R., C.T. (Carlos Toledano) and C.T. (Carlos Torres); data curation, M.S. and C.C.-J.; writing—original draft preparation, M.S.; writing—review and editing, M.S., C.C.-J. and C.G.-D.; supervision, M.S.; funding acquisition, M.S., A.B. and A.C. All authors have read and agreed to the published version of the manuscript.

**Funding:** This research was funded by the Spanish Ministry of Science and Innovation (PID2020-118793GA-I00, PID2019-104205GB-C21, EQC2018-004686-P and PID2019-103886RB-I00), the H2020 program from the European Union (GA no. 19ENV04, 654109, 778349, 871115 and 101008004), and the Unit of Excellence "María de Maeztu" (MDM-2017-0737) financed by the Spanish State Research Agency (AEI). The authors wish to thank ACTRIS, AEROSPAIN and Junta de Castilla y León (ref: VA227P20) for supporting the calibration of the AERONET sun photometers used in this publication, and also to Ayuntamiento de Tazacorte, Ayuntamiento de Fuencaliente and Cabildo Insular de La Palma for their help in terms of infrastructure and logistics. M.-Á.L.-C. and C.V.C.-P. are supported by the INTA predoctoral contract program. E.J.W. is funded by the NASA Radiation Sciences Program and Earth Observing System.

**Data Availability Statement:** The data used in this work are also used in other ongoing studies about the transport of the volcanic plume and are not public at the moment. They can be obtained upon request from the corresponding author.

**Conflicts of Interest:** The authors declare no conflict of interest.

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
