# Peer review of "Volcanic Eruption of Cumbre Vieja, La Palma, Spain: A First Insight to the Particulate Matter Injected in the Troposphere"

_remotesensing, doi:10.3390/rs14102470_

Round 1

Reviewer 1 Report

Dear Authors,

Comments and Suggestions for Authors

The paper by Sicard et al., “Volcanic eruption of Cumbre Vieja, La Palma, Spain: A first insight to the particulate matter injected in the troposphere” provides some results on the characterization of the ash plume due to the eruption of the Cumbre Vieja volcano, which started on 19 September and ended on 13 December 2021.

Paper results are mainly based on observations from a the micro‐pulse lidar system operating at Tazacorte (28.641ºN, 17.933ºW, 140 m asl) and a sun/sky‐photometer, which was operated in Fuencaliente  (21 km southeast of Tazacorte) from 21 September, 2021, until 23 January, 2022. More specifically, the authors focused on the measurements on 18 October and 15 November 2021.

The paper is clearly written, but methodology, discussion, and results are lacking as highlighted in the following, likely because the manuscript is a Technical note and not a Research paper.

Main comments concern the following points:

  • Line 21 The abstract should report main parameters (backscatter coefficient, AOD, volume and particle depolarization ratios) characterizing the ash plume at least during the two investigated case studies on 18 October and 15 November. The43±0.45 km value represents the mean daily value?

  • Satellite images showing the ash plume at least on 18 October and 15 November would be helpful to make the paper self-consistent, even if they can be found on the web as the authors have indicated.

  • Lines 88-89 and 92-93 should be deleted since the weather station data, the SONA all-sky camera (all-sky images in the spectral range 400-700 nm), and the Thermo Teco ozone analyzer (ozone concentration) have not be used to deepen the paper analysis.

  • Lines 99-100. How was detected the height of the peak of the lowermost volcanic plume? The used methodology should be mentioned/described in the manuscript being one of the paper main results and since volcanic ash plumes are generally characterized by a quite wavy structure.

  • Lines 118-130. I believe that the calculation of the mass concentration value, which also represent one of the main paper results, deserves a more general analysis based on previous studies [Scollo et al., 2011; Gasteiger et al., 2011; Revuelta et al., 2012; Pisani et al., 2012; Romano et al., 2018] and not only on [13]. Sensitivity tests should also be provided. Note that Scollo et al.( 2011) and Pisani et al.(2012) characterized fresh volcanic ash from Mt. Etna. Comparisons between paper results and the ones of previous studies mainly based on fresh volcanic ash should also be reported, to properly contribute to the characterization of the fresh ash properties.

  • Figure 2. I suggest to show the volume depolarization values on the right axis of Fig. 2b. The AOD retrieved from the lidar backscatter profile by using S=50 sr in addition to the AOD values from AERONET should be reported on Fig. 2b. Some comments on the differences between AERONET and lidar AODs should be provided. Why the lidar ratio was not calculated by combining lidar and AERONET measurements, as it is commonly done within ACTRIS?

Author Response

Thank you very much. We greatly appreciate the reviewer positive feedback. Our answers are included in the following pdf file.

Reviewer 2 Report

The paper reports original data. Data and methods are well described. Results and discussion are clear and without over interpretation. The paper deserves publication. 

Author Response

Thank you very much. We greatly appreciate the reviewer positive feedback.

Reviewer 3 Report

This manuscript contains a description of a new data set of aerosol measurement during the recent eruption of Cumbre Vieja volcano (La Palma, Spain). It is very well written and I recommend it for pulication. I do have some minor comments for rephrasing a sentence or including another reference (see comments in attached pdf). But as a technical note (as specified in the header of the manuscript) it contains all information needed to describe the data set.

Author Response

(The authors gave the same response as above.)

Round 2

Reviewer 1 Report

I do not have any comment.